# Identifying Genes Related to Acute Myocardial Infarction Based on Network Control Capability

**DOI:** 10.3390/genes13071238

**Published:** 2022-07-13

**Authors:** Yanhui Wang, Huimin Xian

**Affiliations:** College of Mathematics and Systems Science, Shandong University of Science and Technology, Qingdao 266590, China; huiminxian78@163.com

**Keywords:** acute myocardial infarction (AMI), control capability, differential control capability genes (DCCGs)

## Abstract

Identifying genes significantly related to diseases is a focus in the study of disease mechanisms. In this paper, from the perspective of integrated analysis and dynamic control, a method for identifying genes significantly related to diseases based on logic networks constructed by the LAPP method, referred to as NCCM, is proposed and applied to the study of the mechanism of acute myocardial infarction (AMI). It is found that 82.35% of 17 differential control capability genes (DCCGs) identified by NCCM are significantly correlated with AMI/MI in the literature and DISEASES database. The enrichment analysis of DCCGs shows that AMI is closely related to the positive regulation of vascular-associated smooth muscle cell proliferation and regulation of cytokine production involved in the immune response, in which *HBEGF*, *THBS1*, *NR4A3*, *NLRP3*, *EDN1*, and *MMP9* play a crucial role. In addition, although the expression levels of *CNOT6L* and *ACYP1* are not significantly different between the control group and the AMI group, NCCM shows that they are significantly associated with AMI. Although this result still needs further verification, it shows that the method can not only identify genes with large differences in expression but also identify genes that are associated with diseases but with small changes in expression.

## 1. Introduction

Acute myocardial infarction (AMI) is a multifactorial disease, which is myocardial necrosis caused by acute and persistent ischemia and hypoxia of the coronary arteries. It can be complicated by arrhythmia, shock, or heart failure, and is mostly on the basis of atherosclerotic stenosis [1]. The occurrence of AMI is often the result of the interaction between genetics and the environment. It is one of the main causes of death and disability in the world [2]. Therefore, one of the focuses of current research is to identify the mechanism of AMI and clarify how to diagnose, prevent, and treat the disease.

So far, it has been found that AMI leads to cardiomyocyte apoptosis through multiple mechanisms, of which the inflammatory response is crucial in determining the myocardial infarction (MI) size [3]. Individual differences in inflammatory responses [4] and regulation of immune cells [2] can be used as new treatment strategies. Clinical studies have found that monocyte-platelet aggregates, heart fatty-acid-binding protein, cardiac troponin (cTn), and microRNAs (miRNAs) are valuable biomarkers [5,6,7], and cardiac troponin detection can be used for the diagnosis of AMI. Using network reconstruction, bioinformatics, and other methods, it was found from a static perspective that some genes induce AMI by regulating the immune and inflammatory responses and metabolic processes [8,9]. The identification of disease markers is the key to reducing the probability of future disease in terms of risk prediction and potential intervention. Therefore, the identification of genes significantly related to AMI is helpful to study the disease mechanism, prevention, and treatment measures of AMI.

Only the interaction among genes can have functions. So, the genes are regarded as nodes, and the interactions between genes form edges. This leads to a network. Zhang et al. [8] proposed a method to extract key modules related to AMI through a weighted gene co-expression network, but its analysis needs to set multiple thresholds for computing hub genes of modules. Kamran et al. [10] proposed a new measure of the topology properties of the network; however, this measure ignores the relationship between the various nodes. Ding et al. [11] proposed the NIPMI method to quantify the similarity between genes based on the Interaction Part Mutual Information (IPMI) of network construction measurement, which is used to detect characteristic genes from multi-cancer data. Although Ding et al. conducted local and global studies on gene screening, these studies were based on the static parameters of the network. However, biological systems evolve dynamically, and there are few studies on key genes from the perspective of network dynamics. In this paper, we propose a method to identify key nodes in the network from the perspective of network dynamic structure parameters, and then use this method to identify genes significantly related to AMI.

For a finite directed network, if the state of a node is changed when a signal is applied to it, it will affect other nodes. The more nodes it affects, the stronger its control capability in the network, which is called the node control capability [12]. When a node can control all nodes in the network, the network is said to be controllable [13]. If the same node has a different control capability between the disease network and the health network, it means that the structure of the two networks has changed. The greater the difference in the control capability of a node between two networks, the greater the contribution of the node to the network structure difference between the two cases, and thus the node is related to the disease. Based on this idea, a method is proposed to identify disease-related genes based on network control capability, which is called NCCM.

To verify the effectiveness of NCCM, AMI is taken as an example. Based on the gene expression profiling data of AMI, Bowers’ LAPP method [14] was used to construct a first-order logic network (also known as a directed network). Using NCCM, 17 differential control capability genes (DCCGs) related to AMI were identified, 14 of which (*NR4A3* [15], *THBS1* [16,17], *CXCL3* [18], *ITLN1* [19,20], *CLEC4D* [21], *LRG1* [22], *IRAK3* [23,24], *HBEGF* [25], *MMP9* [26,27], *NLRP3* [28,29,30], *EDN1* [31], *VNN3* [24], and *PDK4* [32]) were significantly related to the growth, proliferation, and repair of AMI/MI cells, and *BCL6* [33] was indirectly related. The enrichment analysis of DCCGs showed that AMI was significantly related to the positive regulation of smooth muscle cell proliferation and the regulation of cytokine production involved in the immune response. In addition, the *CNOT6L* and *ACYP1* genes were identified to be related to AMI, which provides a basis for biomedical researchers to further study the pathogenic mechanism, detection, and treatment of AMI.

## 2. Materials and Methods

### 2.1. Study Design

A method is established to identify disease-related genes based on the control capability of logic networks in this paper. Firstly, based on the expression profiles in two states: the control group (ConGroup) and the experiment group (ExpGroup), first-order logic networks are built for the two groups using the LAPP method [14], respectively. Secondly, the control capability of each node in the two networks is calculated and the average control capability of the two networks is used to evaluate the separability of the control capability of these two networks from a statistical point of view. Thirdly, if the curves of the average control capability of the two networks have no intersection, which means they are separated, differential control capability genes (DCCGs) are selected according to the difference in the control ability of the nodes between the two groups. Finally, a four-dimensional analysis of DCCGs is performed: KEGG pathway, GO, comparison with other methods, and disease–gene associations analysis (Figure 1).

### 2.2. Control Capabilities

The study of network controllability mainly focuses on how to select nodes that receive external input, so that the system can reach any desired final state from any initial state in a limited time. Nodes that accept external input are called the control (or input) nodes [34]. The more nodes it affects, the stronger its control capability [12]. When the input nodes can control all the nodes of the network, the network system is said to be controllable [13]. The state change of these control nodes will cause the state change of the whole system, so these control nodes play a vital role in the whole system. The node controllability of the directed networks is briefly recalled below.

Let Γ be a directed network with N nodes. Let $A_{N}=(a_{ij})$ be an adjacency matrix of N×N, where $a_{ij}$ represents the weight of the directed edge from node j to node i; $x={\left(x_{1},x_{2},\ldots ,x_{N}\right)}^{T}$, where $x_{i}(i=1,2,\ldots ,N)$ represents the state of the node i. Let $u={\left(u_{1},u_{2},\ldots ,u_{m}\right)}^{T}$ denote m control input signals, where $u_{k}$ denotes the k-th input signal (k=1,2,…,m); $B=(b_{ik})$ is a 0–1 control matrix of N×m, where if the k-th signal $u_{k}$ controls the i-th node, $b_{ik}=1$; otherwise, $b_{ik}=0$. Then, the ordinary differential equation of the network system Γ is expressed as [12]:(1)$$\dot{x}=Ax+Bu$$

Let $C=(B,AB,A^{2}B,\ldots ,A^{N-1}B)$; then, C is called the controllability matrix. According to the Kalman rank criterion [34], if rankC=N, then system (1) is controllable.

In particular, if only the influence of a certain node i on the whole network is studied, then only this node i in the network can receive the input signal. In this case, all signals only act on node i. For convenience, it may be assumed that there is only one control input signal, and this signal only acts on the node i. So:$$B=[e_{i}]={\left[\begin{array}{ccccccc} 0 & 0 & \ldots & \underset{i}{\underbrace{1}} & \ldots & 0 & 0 \end{array}\right]}^{T}$$

Furthermore, rankC represents the strength of the control capability of node i [12]. If rankC=N, then node i can control the whole system. If rankC<N, then only the rankC-dimensional subspace can be controlled. If rankC=1, then node i can only control itself.

**Definition** **1**[12]**.** *When there is only one control input signal in the directed network*
Γ* and the signal only acts on node *
i*, *
rankC
*is called the control capability of node *
i.

The average value of the control capability of N nodes in network Γ is called the control capability of network Γ.

### 2.3. Construction of Logic Networks

Based on the gene expression profile data, a first-order logic network is constructed according to the LAPP method using MATLAB software (MATLAB v. 7.8, Cleve Moler, USA). The specific process is as follows.

For genes A and B, we regard them as information sources or random variables, and the corresponding data are represented by X and Y:

$X=\left\{x_{1},x_{2},\ldots ,x_{n}\right\}$ with expression probability $p(X=x_{i})(i=1,2,\ldots ,n)$;

$Y=\left\{y_{1},y_{2},\ldots ,y_{n}\right\}$ with expression probability $p(Y=y_{i})(i=1,2,\ldots ,n)$.

Step 1: Data segmentation.

Bowers [14] calculated the weight U of a logic relation between genes based on 0–1 discrete data. However, discretization of continuous data into 0–1 data will lose information, so this paper performs fine segmentation. The gene expression profile data is normalized. Gene A’s data $X=\left\{x_{1},x_{2},\ldots ,x_{n}\right\}$ is taken as an example. It is assumed that $x_{i}\in [0,1]$ for i=1,2,…,n. The [0, 1] is divided into 2k parts, k∈Ν. If $x_{i}<0.5$ and $x_{i}\in [\frac{l}{2k},\frac{l+1}{2k})$, then $x_{i}$ is replaced by $\frac{l}{2k}$; if $x_{i}\geq 0.5$ and $x_{i}\in (\frac{l}{2k},\frac{l+1}{2k}]$, then $x_{i}$ is replaced by $\frac{l+1}{2k}$,where l∈{0,1,2,…,2k−1}.

Step 2: Calculate the information entropy.

$H(X)=E[-\log_{n}p(x_{i})]=-\sum p(x_{i})\log p(x_{i})$, where $X=\left\{x_{1},x_{2},\ldots ,x_{n}\right\}$, $p(x_{i})$ is the expression probability corresponding to X, then H(X) represents the information entropy of gene A.

Step 3: Calculate the joint entropy.

$H(X,Y)=-\sum p(x_{i},y_{i})\log p(x_{i},y_{i})$ is the joint entropy of genes A and B.

Step 4: Calculate mutual information.

I(X,Y)=H(X)+H(Y)−H(X,Y) represents the mutual information of genes A and B.

Step 5: Calculate the U value.

U(X|Y)=I(X,Y)/H(X), where 0≤U(X|Y)≤1 represents the effect on the certainty of X when Y is determined. It represents the possibility of the existence of the uncertain logical relationship “A→B”. Moreover, the closeness of the nodes’ association can be characterized by the U value. The greater the U value of the two nodes, the greater the influence between the nodes.

Step 6: Filter the logical relationship.

It is assumed that the logic direction threshold is α. For genes A and B, x=U(Β|A), y=U(A|Β). If $\left|x-y\right|\leq \alpha$, it means that genes A and B are very close and affect each other, so a logic relation between A and B does not exist. If $\left|x-y\right|>\alpha$, it means that there is a big difference between the influence of A on B and the influence of B on A, and then it is considered that there is a logic relationship between A and B, and further, the larger one of x and y is taken. In this paper, α=0.

### 2.4. Identify DCCGs

If the control capabilities of two networks with the same node are not equal, their structures are different. Statistically, if the control capability curves of two control groups do not intersect within the threshold $\left(t_{1},t_{2}\right)$ $\left(t_{1}<t_{2}\;and\;t_{1},t_{2}\in \left\{0.1,\ldots ,0.9\right\}\right)$, then their control capability curves are said to be separable, which further indicates that there are structural differences in the controllability between the networks of the two control groups. The control capability of a network is determined by the control capability of its nodes. The greater the difference in the controllability of the nodes representing the same object in the two networks, the greater its contribution to the difference in the control capability of the two networks. Therefore, the difference in the control capability of the nodes between the two groups can be used as an indicator of the nodes with significant differences in their network structure.

It is assumed that the curves of the controllability of the logic networks of the two control groups with the threshold $\left(t_{1},t_{2}\right)$ $\left(t_{1}<t_{2}\;and\;t_{1},t_{2}\in \left\{0.1,\ldots ,0.9\right\}\right)$ are separable. A method for identifying genes closely related to diseases is built in terms of the network control capabilities in the following, named the network control capability method, simply denoted by NCCM.

Step 1: R software (R v. 4.0.2, Ross Ihaka and Robert Gentleman, Auckland, New Zealand) is used to calculate the control capability of each node v in the logic network with a step size of 0.1 under the threshold $\left(t_{1},t_{2}\right)$ of two control groups, respectively. Denoted by $\left(v_{ConGroup1},v_{ConGroup2},\ldots ,v_{ConGroup\lambda }\right)$ and $\left(v_{ExpGroup1},v_{ExpGroup2},\ldots ,v_{ExpGroup\lambda }\right)$, respectively, where $\lambda =\frac{t_{2}-t_{1}}{0.1}+1$, $v_{ConGroupi}$ and $v_{ExpGroupi}$ represent the control capabilities of node v in the networks of two control groups under threshold $t_{1}+\frac{i-1}{10}$, i=1,2,…,λ, respectively.

Step 2: The average value $CC_{v}$ of the control capability difference between the logical networks with a step size of 0.1 within the threshold $\left(t_{1},t_{2}\right)$ of each node v in the two control groups is calculated, that is:$$CC_{v}=\frac{\sum_{i=1}^{\lambda }\left|v_{ConGroupi}-v_{ExpGroupi}\right|}{\lambda }$$

Step 3: The threshold of significant difference in control capability β(β>0) is set and the nodes with the significant differences in control capability are filtered. If $CC_{v}>\beta$, the node v is called a node with a significant difference in the control capability between two control groups, and its representative gene is called a gene with difference in control capability, simply denoted by DCCGs.

## 3. Case and Results

### 3.1. Data

GSE66360 in the NCBI database is taken as an example. GSE66360 was obtained from 50 healthy subjects and 49 clinical subjects with AMI [35]. RNA samples were isolated from subjects’ CD146+ cells and processed by Affymetrix human U133 Plus 2.0 array (Affymetrix, Santa Clara, CA, USA), from which two groups of Discovery and Validation were formed [36]: The Discovery group consisted of a dataset of 22 healthy subjects (control group) and 21 acute myocardial infarction patients (AMI group), and the Validation group consisted of a dataset of 28 healthy subjects and 28 AMI patients. In [36], Discovery was taken as the research object and 126 differential expression genes (DEGs) were screened out according to the hypothesis test of gene expression profiles [37] (*p* < 0.0001). This paper still uses these 126 DEGs as the research object and does not carry out any processing of excluded data.

### 3.2. Assessment of DEGs’ Networks

Based on the gene expression profile data of DEGs in AMI, first-order logic networks (take α=0) of the control group and the AMI group are constructed using the LAPP method, and then the control capability of the DEGs and the network control capability of the two groups in the logic networks within a threshold of 0.1–0.9 are calculated. It is found that the control capability curves of the logic networks among the threshold of 0.1–0.7 of the control group and the AMI group are separable (Figure 2a). The network of the AMI group under threshold 0.6 is significantly more complex than that of the control group (Figure 2b); that is, there are more associations. This shows that there is a big difference in the network structure between the control group and the AMI group, and this difference is related to the disease, so it is reasonable to use the network difference to study the mechanism of the disease.

### 3.3. DCCGs Identification

In the logic networks with a separability threshold of 0.1~0.7, there are 59 genes with an average difference in the control capability of nodes of 5 or more. Among them, the expression of 55 genes in the AMI group is higher than that in the control group (Figure 3a), and the control capabilities of 52 genes in the AMI group are higher than that in the control group (Figure 3b). According to the literature analysis of the Web of Science database in the last 10 years, the greater the difference threshold β of the node control capability between the two groups, the higher the proportion of confirmed disease-related genes (Figure 3c). β=10. In total, 17 DCCGs were screened (Table 1). In total, 13 out of 17 DCCGs: *NR4A3*, *THBS1*, *CXCL3*, *ITLN1*, *CLEC4D*, *LRG1*, *IRAK3*, *HBEGF*, *MMP9*, *NLRP3*, *EDN1*, *VNN3*, and *PDK4*, were confirmed to be significantly related with AMI/MI. According to the disease–gene associations Z-score in DISEASES (https://diseases.jensenlab.org/, accessed on 20 December 2021), 76.5% of DCCGs are associated with AMI/MI: *NR4A3*, *THBS1*, *CXCL3*, *BCL6*, *ITLN1*, *LRG1*, *IRAK3*, *HBEGF*, *MMP9*, *NLRP3*, *EDN1*, *VNN3*, and *PDK4*. To sum up, the literature and DISEASES show that 82.35% of DCCGs are related to AMI/MI: *NR4A3*, *THBS1*, *CXCL3*, *BCL6*, *ITLN1*, *CLEC4D*, *LRG1*, *IRAK3*, *HBEGF*, *MMP9*, *NLRP3*, *EDN1*, *VNN3*, and *PDK4*.

### 3.4. DCCGs Analysis

#### 3.4.1. Enrichment Analysis

Gene set enrichment was performed by the Metascape server (https://metascape.org/, accessed on 20 January 2022) [39]. In addition to *AC079305.10*, 16 DCCGs were recognized by the Metascape server. Pathway and process enrichment analysis showed that DCCGs are mainly involved in positive regulation of smooth muscle cell proliferation, and regulation of cytokine production involved in the immune response (Table 2).

The relationships between the enriched terms were further captured using Metascape. A subset of enriched terms was selected and rendered as a network plot, where terms with a similarity > 0.3 are connected by edges (Figure 4a).

The top three genes in the difference of controllability: *HBEGF*, *THBS1*, and *NR4A3*, are shared by the terms GO:0048661, WP2865, and GO:1904707. These functions are mainly positive regulation of vascular-associated smooth muscle cell proliferation and IL1 and megakaryocyte in obesity. *MMP9* and *EDN1*, which have very high MI-related scores in the Diseases database of 5.9 and 6.0, respectively (Table 1), are also involved in the positive regulation of vascular-associated smooth muscle cell proliferation. Figure 4b shows the correlation of other DCCGs with enriched terms.

As is shown in Figure 4c, in the top-level GO biological processes, DCCGs are mainly enriched in GO:0048518, GO:0050789, and GO:0002376, which correspond to positive regulation of biological process, regulation of biological process, and immune system process, respectively.

#### 3.4.2. KEGG Pathway

The DAVID online tool (https://david.ncifcrf.gov/DAVID 6.8, accessed on 13 December 2021) and the KOBAS (http://kobas.cbi.pku.edu.cn/anno_iden.php) were used to analyze DCCGs’ KEGG pathway with *p*-value < 0.05 and adjusted *p* < 0.05 (Figure 5).

*THBS1*, *HBEGF**,* and *MMP9* are jointly involved in bladder cancer. *HBEGF* acts on *MMP9* through the transactivation of epidermal growth factor receptor (EGFR) and ErbB signaling pathway, induces endothelial cell migration through degradation of the extracellular matrix, and finally causes angiogenesis (Figure 5a). *NR4A3* and *MMP9* are involved in transcriptional misregulation in cancer (Figure 5b). Studies have shown that aberrant angiogenic processes are involved in the pathogenesis of diseases such as cancer. Moreover, angiogenesis-related mechanisms can improve tissue regeneration after conditions such as arteriosclerosis and myocardial infarction [40]. An lnc RNA named urothelial carcinoma-associated 1 (UCA1) is found in tumors such as bladder cancer and lung cancer. Furthermore, it is found that UCA1 could be used as a promising novel biomarker for the diagnosis and/or prognosis of AMI [41]. *EDN1* and *MMP9* participate in the TNF signaling pathway while *CXCL3* participates in this signaling pathway together (Figure 5c). In addition, *EDN1* and *MMP9* also jointly participate in the Relaxin signaling pathway (Figure 5d), fluid shear stress, and atherosclerosis (Figure 5e). *EDN1* controls *MMP9* through the PI3K-Akt signaling pathway (Figure 5d) and is upstream of *MMP9* through the MAPK signaling pathway.

#### 3.4.3. Comparison with Other Methods

At present, most methods for screening key nodes in the network are from the perspective of network static structure parameters, and the most widely used methods are based on the centrality of nodes such as the degree centrality, eigenvector centrality, etc. [8,9]. The following is a method for screening differential nodes between two groups based on the degree. This method is abbreviated as DDM here.

Suppose that the curves of the average degree of the networks of two control groups among the threshold $\left(t_{1},t_{2}\right)$ $\left(t_{1}<t_{2}\;and\;t_{1},t_{2}\in \left\{0.1,\ldots ,0.9\right\}\right)$ are disjointed. The specific steps of DDM are as follows.

Step 1: The degree of each node u in the logic network with a step size of 0.1 in the threshold $\left(t_{1},t_{2}\right)$ of two control groups is calculated, respectively, denoted by $\left(u_{ConGroup1},u_{ConGroup2},\ldots ,u_{ConGroup\lambda }\right)$ and $\left(u_{ExpGroup1},u_{ExpGroup2},\ldots ,u_{ExpGroup\lambda }\right)$, respectively, where $\lambda =\frac{t_{2}-t_{1}}{0.1}+1$, $u_{ConGroupi}$, and $u_{ExpGroupi}$ represent the degrees of node u in the networks of two control groups under threshold $t_{1}+\frac{i-1}{10}$, i=1,2,…,λ, respectively.

Step 2: The average value $D_{u}$ of the degree difference between the logic networks with a step size of 0.1 within the threshold $\left(t_{1},t_{2}\right)$ of each node u in the two control groups is calculated; that is:$$D_{u}=\frac{\sum_{i=1}^{\lambda }\left|u_{ConGroupi}-u_{ExpGroupi}\right|}{\lambda }$$

Step 3: The threshold of significant difference in degree γ(γ>0) is set and the nodes with a significant difference in the degree are filtered. If $D_{u}>\gamma$, the node u is called a node with a significant difference in the degree between two control groups, and its representative gene is called a gene with difference in degree, simply denoted as DDGs.

The degree curves of the logic networks of DEGs in the AMI and control groups are separable within thresholds (0.1, 0.9) (Figure 6). Then,γ=30, and 17 DDGs are screened out (Table 3). About 58.8% of DDGs are found to be associated with AMI/MI in the DISEASES database while 76.5% of DCCGs screened by NCCM are associated with AMI/MI. This indicates that NCCM is more effective than DDM in identifying AMI-related genes.

## 4. Discussion

Based on the difference in the control capability of the logic networks between two control groups, a method identifying key nodes in the network was established, named NCCM. AMI was taken as an example. The logic networks of 126 DEGs of the control group and the AMI group were established, and the separability of the curves of the network control capability of the control group and the AMI group was studied. It was found that the curves of the control capability of the logic networks of the two groups under thresholds of 0.1–0.7 do not intersect (Figure 2a). This shows that there are differences in the network structure between the control group and the AMI group, which is caused by AMI disease. Further, according to NCCM, nodes with a control capability difference of more than 10 were selected as the significantly different nodes, and their corresponding genes were DCCGs. In the literature and DISEASES, 82.35% of DCCGs (Table 1) were associated with AMI/MI. Compared with DDGS screened by DDM, NCCM is more effective. Through the enrichment analysis of DCCGs, it was found that AMI disease is significantly related to the positive regulation of vascular-related smooth muscle cell proliferation, and *HBEGF*, *THBS1*, *NR4A3*, *NLRP3*, *EDN1*, and *MMP9*, which are involved in this function, are significantly related to AMI/MI in DISEASES and the literature.

*HBEGF* is necessary for maintaining normal function of the adult heart [42]. *HBEGF* is known to induce cell growth in various cell types via transactivation of epidermal growth factor receptor (EGFR). Tanaka et al. suggested the interaction between *HBEGF* and EGFR transactivation is closely related to the proliferation of cardiac fibroblasts and cardiac remodeling after MI in an autocrine, paracrine, and juxtacrine manner [43]. Mouton et al. studied in vitro day 7 MI fibroblast secretome-repressed angiogenesis through *THBS1* signaling [16]. *NR4A3* clusters are novel functional modules in the CD146+ cell-mediated immune-inflammatory balance, triggering increased susceptibility to vascular deterioration and accelerating myocardial injury [16]. *NR4A3* could inhibit post-AMI inflammation responses via JAK2-STAT3/NF-kappa B signaling and may well be a therapeutic target for cardiac remodeling after AMI [15]. *NLRP3* (cryopyrin) is central in AMI: it reduces apoptosis, infarct size, and cardiac dysfunction during AMI [28]. Ablation of *MMP9* decreases infarct size in the non-diabetic myocardial infarction heart [27]. The rs3918242 polymorphism of the *MMP9* gene plays a primary role in the risk of developing MI [44]. *EDN1* induces CDH2 and VEGF expression in hUCB-MSCs, leading to improved therapeutic efficacy in rat MI, suggesting that *EDN1* is a potential priming agent for MSCs in regenerative medicine [31]. In Table 2, *HBEGF*, *THBS1*, *NR4A3*, *NLRP3*, *EDN1*, and *MMP9* are shared in terms of positive regulation of vascular-associated smooth muscle cell proliferation. Zhang et al. proposed that high-glucose-induced proliferation of vascular smooth muscle cells (VSMCs) plays an important role in the development of diabetic vascular diseases. Interferon regulatory factor-1 (Irf-1) is a positive regulator of the high-glucose-induced proliferation of VSMCs. However, the mechanisms remain to be determined [45].

In String (https://cn.string-db.org/, accessed on 13 February 2022) [46], when the threshold of a correlation coefficient is 0.4, *HBEGF*, *THBS1*, *NR4A3*, *NLRP3*, *EDN1*, and *MMP9* are related (Figure 7a). In addition, when the threshold of a correlation coefficient is 0.15, *ACYP1* interacts with *THBS1* and *NR4A3*, but there is no association between *CNOT6L* and other genes (Figure 7b).

Both the CT and CTsv products of the *ACYP1* gene were able to induce a proapoptotic effect when expressed in the HeLa cell line [47]. *CNOT6L* is a validated target of miR-146a [48]. Exosomes derived from miR-146a-modified adipose-derived stem cells can downregulate early growth response factor 1 to attenuate AMI-induced myocardial damage [49]. Therefore, we predict that *ACYP1* and *CNOT6L* are related to AMI. In the control group and AMI group, the difference in the gene expression between *ACYP1* and *CNOT6L* is low, but the difference in the control capability is high. In this way, from the perspective of gene expression difference, the expression difference of these two genes in the two groups is not significant, they are generally not considered to be related to AMI. However, there are significant differences in the control capability of *ACYP1* and *CNOT6L* between the two groups, so they are significantly correlated with AMI according to NCCM. This shows that the NCCM method can not only identify genes with significant changes in expression but also be effective for genes with weak changes in expression. At present, the mechanism of *ACYP1* and *CNOT6L* in AMI is not clear, which provides a new idea for studying the occurrence and molecular mechanism of AMI.

## 5. Conclusions

From the perspective of the network node control capability, a method identifying key nodes between two networks with the same set of nodes was established, called NCCM. Taking AMI as an example, 82.35% of DCCGs screened by NCCM were related to AMI, which shows that this method is effective in mining disease-related genes. The analysis of DCCGs indicates that AMI is closely related to the positive regulation of vascular smooth muscle cell proliferation and the function of the regulation of cytokine production involved in the immune response. *HBEGF*, *THBS1*, *NR4A3*, *NLRP3*, *EDN1*, and *MMP9* are involved in these functions, and they play an important role in the mechanism of AMI. In addition, it was predicted that *CNOT6L* and *ACYP1*, which have little difference in expression between the control group and the AMI group, are significantly correlated with AMI. This shows that this method can not only identify genes with large differences in expression but also identify genes closely related to AMI but with small changes in expression. This method has certain universality and can also be applied to other data.

## Figures and Tables

**Figure 1 genes-13-01238-f001:**
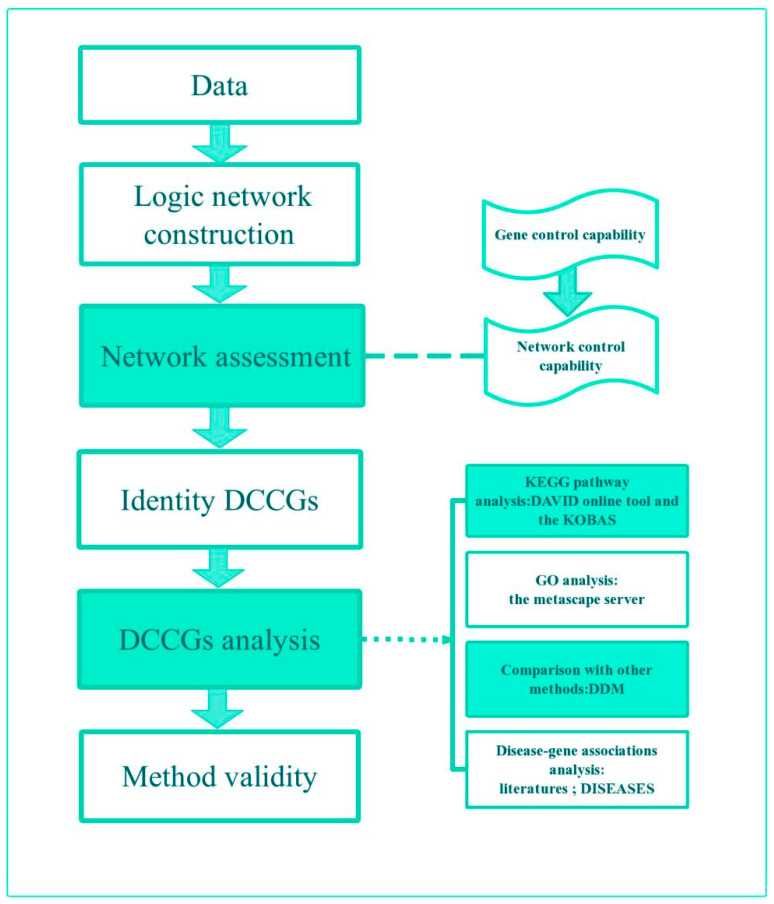
Flowchart for study methods. DCCGs differential control capability genes. DDM method for identifying differential nodes based on the degree.

**Figure 2 genes-13-01238-f002:**
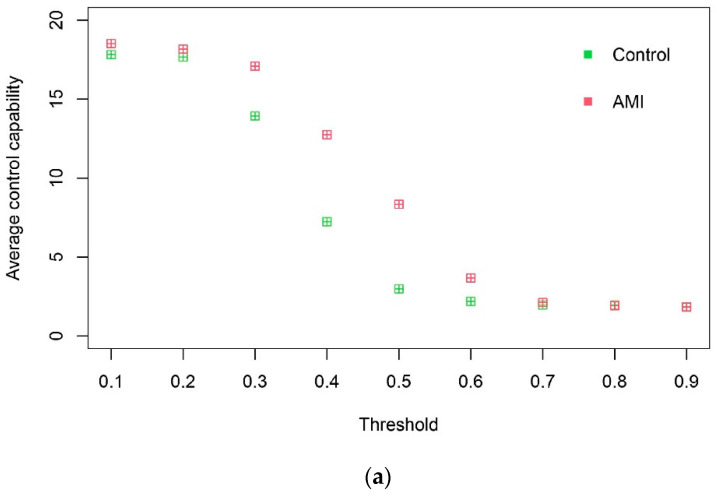
(**a**) Network assessment. Networks in the control and AMI groups are separable according to the average control capability of DEGs; (**b**) logic networks are presented under a threshold of 0.6. The logic networks of DEGs in the AMI group are more complex, compared to that in the control group. Genes represented by node numbers (Appendix A).

**Figure 3 genes-13-01238-f003:**
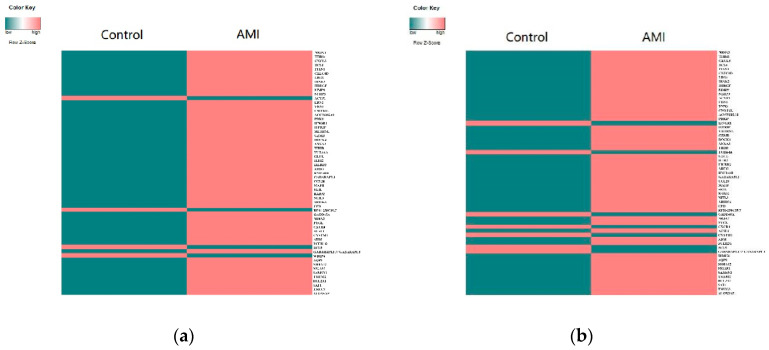
(**a**) Gene expression; (**b**) gene control capability; (**c**) proportion of genes with a control capability above β that have been confirmed to be directly related to AMI/MI.

**Figure 4 genes-13-01238-f004:**
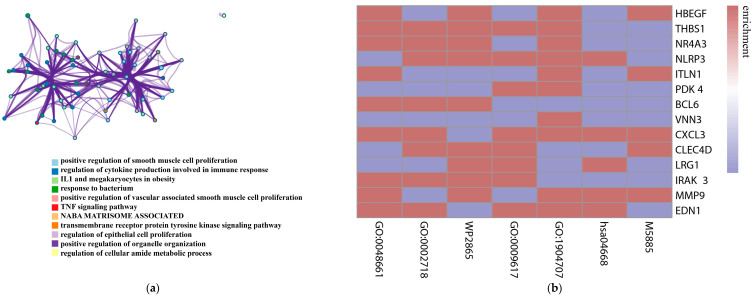
Enrichment of DCCGs. (**a**) Network of enriched terms: colored by cluster ID, where nodes that share the same cluster ID are typically close to each other. (**b**) Enrichment map of genes, where red indicates functional enrichment of genes and blue represents without enrichment. (**c**) The top-level GO biological processes of DCCGs.

**Figure 5 genes-13-01238-f005:**
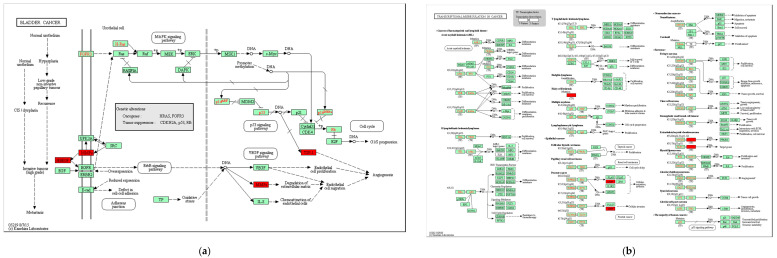
The Kyoto Encyclopedia of Genes and Genomes pathway of differentially control capability genes. Red diamonds represent the DCCGs in AMI. (**a**) Bladder cancer. (**b**) Transcriptional misregulation in cancer. (**c**) TNF signaling pathway involved in *MMP9*, *EDN1*, and *CXCL3*. (**d**) Relaxin signaling pathway. (**e**) Fluid shear stress and atherosclerosis.

**Figure 6 genes-13-01238-f006:**
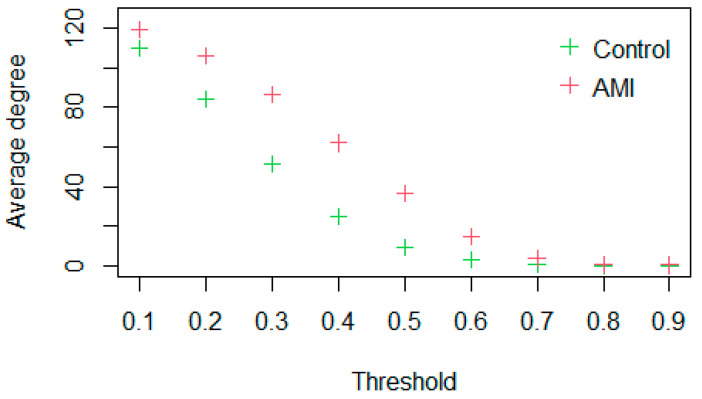
Network assessment. Networks in the control and AMI groups are separable according to the average degree of DEGs.

**Figure 7 genes-13-01238-f007:**
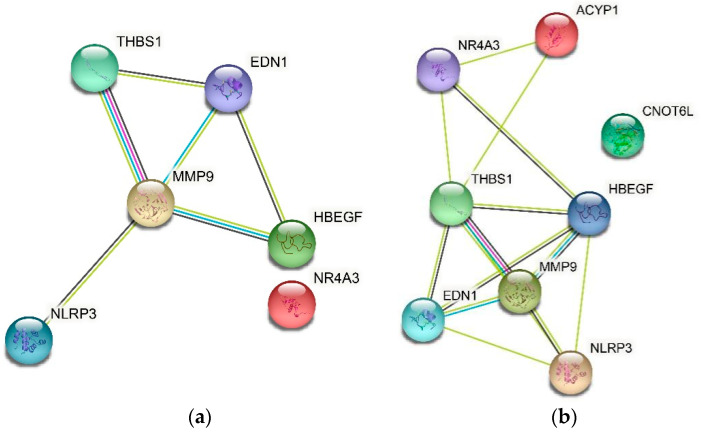
(**a**) *HBEGF*, *THBS1*, *NR4A3*, *NLRP3*, *EDN1*, *MMP9* interaction network at the 0.4 threshold. (**b**) *HBEGF*, *THBS1*, *NR4A3*, *NLRP3*, *EDN1*, *MMP9*, *ACYP1*, *CNOT6L* interaction network at the 0.15 threshold.

**Table 1 genes-13-01238-t001:** Category of DCCGs.

Genes	Gene Names	Control Capability Fold-Change	Gene Expression Fold-Change	Functions in AMI/MI	AMI Z-Score	MI Z-Score
CA^high^GeExp^high^						
*HBEGF*	Heparin-binding EGF-like growth factor	11.5	0.215	Upregulated *HBEGF* plays a pathophysiological role in injured hearts after MI [25].	2.1	3.4
*THBS1*	thrombospondin 1	8.714	0.395	The development of heart failure after acute STEMI [17]; MI fibroblast secretome repressed angiogenesis through *THBS1* signaling [16].	3.1	4.4
*NR4A3*	nuclear receptor subfamily 4 group A member 3	6.684	0.422	Inhibiting post-AMI inflammation responses via JAK2-STAT3/NF-kappa B signaling may well be a therapeutic target for cardiac remodeling after AMI [15].	1.2	2
*BCL6*	BCL6 transcription repressor	4.808	0.239	——	1.2	1.9
*NLRP3*	NLR family pyrin domain-containing 3	3.216	0.411	*RBP4* as a novel modulator promoting cardiomyocyte pyroptosis via interaction with *NLRP3* in AMI [30]; *NLRP3* deletion reduces infarct size during AMI [28]; *NLRP3* inflammasome is upregulated in myocardial fibroblasts post-MI [29].	3.7	4.7
*ITLN1*	intelectin 1	2.735	0.694	The suppression of inflammation in the 6-month post-AMI period might have mediated the significant upregulation of omentin-1, implicating a novel target of treatment [19].	2.4	3.1
*PDK4*	pyruvate dehydrogenase kinase 4	2.375	0.348	Following myocardial infarction, inducible deletion of *PDK4* improved left ventricular function and decreased remodeling [38].	1.9	3.4
CA^high^GeExp^low^						
*ACYP1*	acylphosphatase 1	4.778	−0.071	——		
*CNOT6L*	CCR4-NOT transcription complex subunit 6 like	3.905	0.095	——		
CA^low^GeExp^high^						
*VNN3*	vanin 3, pseudogene	1.833	0.332	Diagnostic biomarkers for STEMI [24].	——	1.3
*CXCL3*	C-X-C motif chemokine ligand 3	1.781	0.412	Associated with reparative phases (post MI) [18].	1.1	2.6
*CLEC4D*	C-type lectin domain family 4 member D	1.691	0.545	Playing an important role in the occurrence and progression AMI [21].		
*LRG1*	Leucine-rich α-2-glycoprotein 1	1.630	0.344	*LRG1*/HIF-1 α promoted H9c2 cell apoptosis and autophagy in hypoxia, potentially providing new ideas for the determination and treatment of AMI [22].	1.3	2.2
*IRAK3*	interleukin 1 receptor-associated kinase 3	1.607	0.456	Silencing of *IRAK3* inactivates the NF-B signaling pathway and prevents AMI progression [23].	2.2	2.8
*MMP9*	matrix metallopeptidase 9	1.575	0.377	Inhibiting the chemokine signaling pathway and leukocyte transendothelial migration play a protective effect on AMI [26]. *MMP9* is upregulated in the diabetic heart, and ablation of *MMP9* decreases the infarct size in the non-diabetic myocardial infarction heart [27].	5	5.9
*EDN1*	endothelin 1	1.474	0.296	*EDN1* induces CDH2 and VEGF expression in hUCB-MSCs, leading to improved therapeutic efficacy in rat MI [31].	5.1	6.0
*AC079305.10*	unnamed	1.389	0.417	——		

DCCGs differential control capability genes; CA control capability; GeExp gene expression; here, the control capability fold-change means AMI/Control. Gene expression fold-change means (AMI-Control)/Control. High expression changes (fold-change > 0.2) with high control capability changes (fold-change > 2), defined as CA^high^GeExp^high^ genes. —— means there is no direct literature to support it. The disease–gene associations Z-score were derived from automatic text mining of the biomedical literature (https://diseases.jensenlab.org/, accessed on 20 December 2021). The higher the Z-score, the better the correlation with it and the higher the trust.

**Table 2 genes-13-01238-t002:** Top seven gene enrichment outputs of DCCGs.

Term	Description	LogP	Genes
GO:0048661	positive regulation of smooth muscle cell proliferation	−8.93	*HBEGF*, *EDN1*, *MMP9*, *THBS1*, *NR4A3*, *BCL6*, *ITLN1*, *IRAK3*, *CXCL3*
GO:0002718	regulation of cytokine production involved in immune response	−6.74	*BCL6*, *NR4A3*, *IRAK3*, *NLRP3*, *THBS1*, *CLEC4D*, *CXCL3*, *EDN1*
WP2865	IL1 and megakaryocytes in obesity	−6.56	*HBEGF*, *MMP9*, *NLRP3*, *BCL6*, *THBS1*, *IRAK3*, *NR4A3*, *CLEC4D*, *LRG1*
GO:0009617	response to bacterium	−5.91	*EDN1*, *CXCL3*, *IRAK3*, *NLRP3*, *LRG1*, *CLEC4D*, *THBS1*, *PDK4*
GO:1904707	positive regulation of vascular associated smooth muscle cell proliferation	−5.74	*EDN1*, *CXCL3*, *IRAK3*, *NLRP3*, *LRG1*, *CLEC4D*, *THBS1*, *PDK4*
hsa04668	TNF signaling pathway	−4.57	*EDN1*, *CXCL3*, *MMP9*, *NLRP3*, *LRG1*
M5885	NABA MATRISOME ASSOCIATED	−4.49	*HBEGF*, *CXCL3*, *MMP9*, *ITLN1*, *CLEC4D*

**Table 3 genes-13-01238-t003:** Category of DDGs.

Genes	Gene Names	AMI Z-Score	MI Z-Score
** *IRAK3* **	interleukin 1 receptor-associated kinase 3	2.2	2.8
** *ITLN1* **	intelectin 1	2.4	3.1
** *BCL6* **	BCL6 transcription repressor	1.2	1.9
** *CXCL3* **	C-X-C motif chemokine ligand 3	1.1	2.6
** *NR4A3* **	nuclear receptor subfamily 4 group A member 3	1.2	2
** *CLEC4D* **	C-type lectin domain family 4 member D	——	——
*CDC25B*	cell division cycle 25B	——	——
** *AC079305.10* **	unnamed	——	——
** *MMP9* **	matrix metallopeptidase 9	5	5.9
*GLUL*	glutamate-ammonia ligase	——	1.9
*FITM2*	fat storage-inducing transmembrane protein 2	——	——
*ITPRIP*	inositol 1,4,5-trisphosphate receptor-interacting protein	——	——
*METRNL*	meteorin-like, glial cell differentiation regulator	——	——
*GABARAPL1*	GABA type A receptor-associated protein-like 1	1.5	2.9
*RNF144B*	ring finger protein 144B	——	——
** *NLRP3* **	NLR family pyrin domain-containing 3	3.7	3.7
*ANXA3*	annexin A3	2.1	2.9

—— Indicates not scored in DISEASES. Bold indicates the same genes as those screened in NCCM.

## Data Availability

Publicly available data sets were analyzed in this study. This data can be found here: https://www.ncbi.nlm.nih.gov (NCBI, accessed on 25 March 2019).

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
