# Peer review of "Identifying Genes Related to Acute Myocardial Infarction Based on Network Control Capability"

_genes, 2022, doi:10.3390/genes13071238_

Round 1

Reviewer 1 Report

Comments and Suggestions for Authors

In the manuscript entitled Identifying genes related to acute myocardial infarction based on network control capability, the authors investigated the genes significantly related to diseases using logic networks constructed using the LAPP method, referred to as NCCM. However, some shortcomings should be resolved before recommending this article for publication.

1.     In the abstract, CNOT6L and ACYP1 are not significantly different between the control and AMI groups. However, NCCM shows that they are significantly associated with AMI. Why are these two genes mentioned, but they are characterized as CAhighGeExplow? I am also concerned with the very low threshold correlation coefficient of ACYP1, which was 0.15 and no association between CNOT6L and other genes using STRING. Why?    

2.     What is the importance of this NCCM approach? Are there any tools that can be used to identify disease-related genes based on network studies? What is the limitation of those tools as compared to your approach?

3.     How can this tool be used for public use? This project needs to be published on GitHub.

4.     Check typos in the whole manuscript.

5.     Make sure the gene names in the whole manuscript are italicized

6.     What is the purpose of constructing separate networks, i.e., control and the AMI group?

7.     Most of the figures are not clear, especially in Fig2. (b). What do the numbers of the nodes refer to? Fig2.(a) and (b) are also not clear.

8.     The format for the whole article is not following the MDPI format.

Reviewer 2 Report

Yanhui Wang et. al. performed an innovative study on identifying acute myocardial infarction (AMI) related genes by bioinformatic analysis. The author proposed the LAPP method and applied in the investigation of AMI. Subsequently, the authors found that genes as HBEGF, THBS1, NR4A3 and etc. were correlated with the AMI disease. This is a well-designed study with some interesting data. The review would like to suggest to publish this article with minor revision. Here are the comments from the reviewer:

1.    It is highly advised that the authors supplement the GO analysis data in the main text of this article.

2.    This article should be extensively revised by a native speaker.

Round 2

Reviewer 1 Report

Most of the comments and suggestions have been addressed. Thank you. But please revise the database name, String into STRING and one more Figure that is not clear, which are Fig. 3(a) and (b). This is because I cannot see the gene names at all.